# Post-Translational Modification-Dependent Activity of Matrix Metalloproteinases

**DOI:** 10.3390/ijms20123077

**Published:** 2019-06-24

**Authors:** Elizabeta Madzharova, Philipp Kastl, Fabio Sabino, Ulrich auf dem Keller

**Affiliations:** Department of Biotechnology and Biomedicine, Technical University of Denmark, DK-2800 Kongens Lyngby, Denmark; elimad@dtu.dk (E.M.); phikas@dtu.dk (P.K.); famisa@dtu.dk (F.S.)

**Keywords:** MMPs, PTMs, glycosylation, phosphorylation, glycosaminoglycans

## Abstract

Due to their capacity to process different proteins of the extracellular matrix (ECM), matrix metalloproteinases (MMPs) were initially described as a family of secreted proteases, functioning as main ECM regulators. However, through proteolytic processing of various biomolecules, MMPs also modulate intra- and extracellular pathways and networks. Thereby, they are functionally implicated in the regulation of multiple physiological and pathological processes. Consequently, MMP activity is tightly regulated through a combination of epigenetic, transcriptional, and post-transcriptional control of gene expression, proteolytic activation, post-translational modifications (PTMs), and extracellular inhibition. In addition, MMPs, their substrates and ECM binding partners are frequently modified by PTMs, which suggests an important role of PTMs in modulating the pleiotropic activities of these proteases. This review summarizes the recent progress towards understanding the role of PTMs (glycosylation, phosphorylation, glycosaminoglycans) on the activity of several members of the MMP family.

## 1. MMP Domain Structure and Classification

Matrix metalloproteinases (MMPs) comprise a family of 23 distinct secreted or membrane-anchored endopeptidases in humans that belong to the metzincin superfamily of metalloproteases. MMPs were initially described as regulators of the extracellular matrix due to their capacity to degrade ECM proteins like collagen, gelatin, laminin, aggrecan, fibronectin, elastin, and proteoglycans. Based on structure and substrate specificity, the MMPs are divided into collagenases (MMP1, MMP8, MMP13), gelatinases (MMP2, MMP9), stromelysins (MMP3, MMP10, MMP11), matrilysins (MMP7, MMP26), membrane-type MMPs (MMP14, MMP15, MMP16, MMP17, MMP24, MMP25), and other non-classified family members (MMP12, MMP19, MMP20, MMP21, MMP23, MMP27, MMP28). MMPs share a multidomain structure consisting of signal peptide, zymogenic pro-peptide domain, a catalytic domain, a variable linker ‘hinge’ region and a hemopexin domain. Some MMPs show variation in the domain arrangements, e.g., MMP7 and MMP26 are lacking the linker and hemopexin domain, MMP23 has a unique cysteine-rich and immunoglobulin-like instead of a hemopexin domain, and the two gelatinases MMP2 and MMP9 have additional fibronectin type-II-related domains. Membrane-type MMPs are anchored to the membrane either via glycosylphosphatidylinositol (GPI) or with help of a transmembrane domain (Figure 1). Additionally, MMPs are distinguished by the highly conserved HExGHxxGxxH motif in the catalytic domain, which contains three histidines that coordinate the zinc molecule in the active site, and the PRCGxPD motif identified in the pro-domain whose cysteine residue coordinates with the active zinc molecule to inhibit proteolysis. Each of these domains is associated with a specific function and very frequently post-translationally modified. This suggests an important role of PTMs in modulating the pleiotropic activities of MMPs [1,2].

## 2. MMP Substrates and Function

MMPs are primarily extracellular proteases, supporting the initial concept that they are generally associated with degradation and regulation of the ECM, thereby influencing many fundamental cellular events involving ECM remodeling [2,3,4,5]. More recently, it was observed that they also can irreversibly process growth factors, cell-surface receptors, cytokines, and chemokines, as well as other MMPs, other proteases and protease inhibitors and even act inside the cell [6]. The activation or inactivation of bioactive molecules by MMP proteolytic processing unraveled unexpected roles of these proteases in the regulation of extra- and intracellular signaling pathways [4,7,8,9]. Hence, the field of MMP research progressed from perceiving these enzymes solely as regulators of ECM to the conception that MMPs are functionally implicated in the regulation of multiple physiological and pathological signaling processes [8,10]. Accordingly, the physiological relevance of their function is emphasized by a direct association between altered expression and/or dysregulation of MMPs and development of pathological conditions, such as chronic inflammatory diseases, vascular diseases, neurological disorders, and cancer [10,11].

## 3. Multilayered Regulation of MMP Activity

Since MMPs regulate major physiological processes, a strict spatiotemporal control of their activity is essential to avoid possible detrimental activities of these proteases. Primarily, expression of MMPs is regulated at the transcriptional level, keeping these enzymes at very low levels in normal tissue homeostasis. MMPs share cis-regulatory elements in their promoter sequences, which allow induction of their expression by stimuli—e.g., in the form of growth factors, cytokines, or hormones. The cooperation between these cis-regulatory elements, coupled with the integration of multiple signaling pathways, provides a wide range of potential interactions between transcriptional regulators, ensuring tissue-specific expression of diverse MMP family members and facilitating a strict control of MMP transcriptional activity. In addition, transcription of MMP encoding genes is regulated by epigenetic mechanisms, such as DNA methylation or histone acetylation [4,5,11,12]. At the post-transcriptional level, regulation of MMP expression is mediated by modulation of mRNA stability and miRNA-based mechanisms that interfere with MMP expression by either transcriptional inhibition or mRNA degradation [13,14,15,16]. An important level of MMP regulation is achieved at the post-translational level, since most MMPs are secreted as inactive pro-enzymes. The ‘cysteine-switch’ region in the pro-peptide domain shields the active site, thereby preventing substrate access and maintaining the enzyme in an inactive state. The activation of proMMPs occurs upon direct proteolytic cleavage of the pro-domain (within the secretory pathway or extracellularly), or induction of conformational changes that disrupt the chelating cysteine residue and enable auto-proteolysis for removal of the pro-domain region [9,12]. Moreover, allosteric activation of proMMPs can be triggered by interaction with ECM components and cell surface molecules and by low-density lipoprotein receptor-related protein (LRP1)-mediated endocytosis [12,17,18,19,20]. Upon activation, mature MMPs are exclusively controlled by endogenous inhibitors, such as tissue inhibitors of metalloproteases (TIMPs) and α_2_-macroglobulin [21,22].

## 4. PTMs—An Additional Level of Protein Regulation

The enormous functional complexity of the cellular proteome is regulated by diverse mechanisms, including transcription, alternative splicing, translation, and PTMs [23]. Among these regulatory mechanisms, PTMs provide a significant genome-independent expansion and diversification of the proteome, thereby creating a continuously fine-tuned regulatory network implicated in many cellular processes [24,25,26]. Based on the type of modification, PTMs can be assigned to several categories: chemical modifications, including methylation, phosphorylation, acetylation, and oxidation; polypeptide modifications, including ubiquitination, SUMOylation and other ubiquitin-like protein conjugation; modifications by complex molecules, including glycosylation, lipids (e.g., acylation, prenylation) and extended structures (e.g., glycosaminoglycans (GAGs)); and modifications of the amino acids or of the polypeptide backbone, including deamidation, eliminylation, and protein cleavage through proteolysis [27,28,29,30]. By reversible or irreversible addition of these functional groups PTMs can modify protein function by altering protein structure, subcellular localization, protein–protein interactions, and degradation, thereby influencing many cellular processes in health and disease [31,32,33,34,35,36].

Like many enzymes, MMPs as well as their substrates are modified by a variety of PTMs. Different domains of the MMP structure are associated with specific functions and undergo different, highly specific PTM modifications, which suggests an important role of PTMs in modulating the pleiotropic activities of MMPs (Figure 2) [1]. Understanding the influence of specific PTMs on the activity of MMPs is necessary to fully understand MMP regulation. In this review, we provide an overview of different PTMs with a focus on glycosylation, phosphorylation, and interaction with extracellular GAGs, and describe their effects on activity of various MMP family members.

### 4.1. Glycosylation of MMP

Glycosylation describes the enzymatic reaction that links saccharides to lipids, other saccharides or proteins and represents the most abundant and diverse PTM. The most common forms of protein glycosylation are the *N*- and *O*-linked glycosylation [37,38]. *N*-linked glycosylation is a glycosidic linkage of glycans to the side chains of asparagine (Asn) residues. It is initiated by a single oligosaccharyltransferase complex with a transfer of *N*-acetylglucosamine (GlcNAc) phosphate at the consensus Asn-Xaa-Ser/Thr sequence (X, indicating any amino acid excluding proline) on the cytosolic side of the endoplasmic reticulum (ER) and extensively modified further in the lumen of the ER and Golgi prior to sorting to secretory vesicles [37,39,40,41]. *O*-linked glycosylation is a glycosidic linkage of glycans to the side chains of serine/threonine (Ser/Thr) or tyrosine (Tyr) residues. *O*-linked glycosylation shows a higher diversity than *N*-glycans, resulting from the complex synthesis of *N*-acetylgalactosamine (GalNAc)-type *O*-linked glycans in the Golgi, which can be initiated by up to 20 different polypeptide GalNAc-transferase isoforms that may compete for the same glycosylation site and the same glycan. Upon transfer of a GalNAc moiety from a donor substrate uridine diphosphate (UDP)-GalNAc to Ser/Thr residues, the *O*-linked glycan is branched out by different core-forming enzymes and further extended by fucosylation and sialylation to the final *O*-linked glycan structure [41,42,43,44,45]. Protein glycosylation regulates essential biological processes, such as protein folding, secretion, cell adhesion, and inter- and intra-cellular trafficking [46,47,48,49,50]. Consequently, alterations in glycosylation patterns are often associated with different pathological conditions like neurodegenerative diseases, diabetes, inflammatory conditions, and cancer [51,52]. The majority of MMPs are glycosylated, and *N*- and *O*-linked glycosylation are present across the MMP family [39]. The following section summarizes the current structural and functional information about glycosylation of different MMPs.

#### 4.1.1. MMP9

MMP9 is the most extensively glycosylated MMP. This protease contains two *N*-linked glycosylation sites, Asn^38^ and Asn^120^ (asparagine residues 38 and 120), in the pro-domain and the catalytic domain, respectively [53]. These *N*-linked glycans have been described as core-fucosylated biantennary structures, partially sialylated with variable fucosylation branches [54]. Despite their similar composition, the individual functions of these *N*-linked glycosylations are distinct, primarily due to their location within the protease. Since *N*-linked glycosylation is generally required for protein secretion, initially a potential role of these two glycans was related to the secretion of MMP9 [55]. Indeed, abrogation of MMP9 glycosylation at Asn^120^ reduced the efficiency of its secretion by increasing the interaction between MMP9 and calreticulin (a protein that prevents misfolded proteins from entering the secretory pathway), thereby inducing MMP9 retention in the ER. The glycosylation at Asn^38^ does not influence the secretion of MMP9, but Asn^38^-glycosylation-deficient MMP9 shows strong amino-acid dependency towards interaction with calreticulin, likely affecting MMP9 secretion in an *N*-glycosylation-independent manner [56]. Additionally, *N*-glycosylation at Asn^38^ was commonly associated with the activation of proMMP9, yet the process of MMP9 activation occurs independently of the glycosylation at Asn^38^. Recent molecular dynamics simulations suggest that the glycosylation at Asn^38^ is indirectly involved in the activation of proMMP9 by inducing conformational changes within the pro-domain, enabling MMP3 to access the two cleavage sites for proteolytic activation [57]. Moreover, interactions between the glycosylation at Asn^38^ and galectin 3 decrease the proteolytic activation of MMP9, whereas interactions with galectin 8 enhance the MMP3-mediated processing, suggesting that the presence of *N*-glycosylation is important for a fine-tuned regulation of MMP9 activity [58,59]. Furthermore, MMP9 has a proline-rich linker sequence between the active site and the hemopexin domain, which contains 14 *O*-linked glycans (denominated OG domain) [60]. Based on their glycan composition, these *O*-linked glycans comprise a heterogenic mixture, varying from core-1 (Galβ1-3GalNAc) to core-2 (Gal-(GlcNAc-)GalNAc) structures and further elongated to larger glycans [61]. The presence of these *O*-linked glycans increases the domain flexibility of MMP9, allowing the protease to adopt multiple enzyme conformations and facilitating individual movements of the catalytic and the hemopexin domain. This influences the recognition, binding, and processing of substrates, cell receptors, and endogenous inhibitors [62,63]. A study by den Steen et al. described that the OG domain is indispensable for correct orientation of the hemopexin domain for MMP9 internalization and degradation by LRP-1 and LRP-2, as well as inhibition by endogenous TIMP1 [60]. Consequently, deletion of this linker region significantly reduced the affinity towards TIMP1 and disrupted the interactions with LRP-1 and LRP-2 [60]. However, since deletion of the OG domain did not affect the activity of MMP9, the authors reported that the OG domain functions as a regulator of extracellular bioavailability of the protease, rather than as a regulator of its activity [60]. An independent analysis of MMP9 lacking the OG domain performed by Vandooren et al. showed a reduced gelatinolytic activity of the truncated protease [64]. Furthermore, a study by Dufour et al. reported reduced MMP9-induced cell migration in endothelial cells expressing MMP9 without the OG domain, which was independent of the proteolytic activity of the enzyme [26].

#### 4.1.2. MMP14

Similar to MMP9, the transmembrane collagenase MMP14 (MT1-MMP) contains *O*-linked glycosylation sites (Thr^291^, Thr^299^, Thr^300^, and Ser^301^) identified in the linker region. The presence of these *O*-linked glycans does not affect zymogen activation, collagenase activity or the autolytic processing of MMP14. However, the *O*-linked glycosylation is important for the formation of a stable complex between MMP14, TIMP2, and proMMP2, which is required for subsequent cell-surface activation of MMP2. Being unable to recruit TIMP2, which connects the catalytic domain of MMP14 and the hemopexin domain of proMMP2, the glycan-deficient MMP14 cannot present a stable trimeric complex on the cell surface and thereby activate MMP2 [65,66,67]. Interestingly, although it has not appeared that glycosylation could directly affect MMP14 activity, recent reports describe increase of MMP14 activity upon alterations in the glycosylation pattern. Indeed, hyper-glycosylation of MMP14 led to higher proteolytic activity and promoted tumor growth [68].

#### 4.1.3. MMP1

MMP1 has two potential glycosylation sites in the active site, but only *N*-glycosylation at Asn^120^ has been experimentally confirmed. Comparison between the glycosylated and non-glycosylated MMP1 showed no significant differences in activity, substrate specificity or inhibitory profiles of the two proteoforms. However, the presence of specific glycan motifs (e.g., α1,3-fucosylated LacdiNAc) can initiate MMP1 selectin-mediated binding to the surface of activated cells through a selectin/glycan interface and therefore may have a profound effect on cell migration [39,69].

#### 4.1.4. MMP2

MMP2 has two potential *N*-linked glycosylation sites, Asn^573^ and Asn^642^, in the hemopexin domain [70,71]. The function of these *N*-linked oligosaccharides remains unclear. However, there are strong implications regarding their involvement in the regulation of MMP2, since the hemopexin domain is involved in MMP2 activation/inhibition, localization of its catalytic activity, and induction of cell signaling upon interaction of MMP2 with cell-surface receptors [71,72].

#### 4.1.5. MMP3

Based on the consensus sequence for *N*-linked glycosylation, MMP3 also has two potential *N*-linked glycosylation sites, Asn^120^ in the catalytic domain and Asn^398^ in the hemopexin domain. However, only a small portion of MMP3 (~20%) is glycosylated, and no correlation has been established between these glycans and the function of the protease [39,73].

#### 4.1.6. MMP13

MMP13 was shown to be potentially glycosylated at two asparagine residues (Asn^117^ and Asn^152^) in the catalytic domain. The *N*-linked glycosylation site at Asn^117^ has been experimentally verified, but the function of this *N*-linked glycan has not been determined since no differences were observed between glycosylated and non-glycosylated recombinant MMP13 [74,75].

#### 4.1.7. MMP17

MMP17 (MT4-MMP), a glycosylphosphatidylinositol-anchored matrix metalloproteinase, has two *N*-linked glycans at Asn^137^ and Asn^318^ in the catalytic site and the linker region, respectively. The presence of these *N*-linked oligosaccharides stabilizes the dimeric form of MMP17 by promoting non-covalent interactions or facilitating folding and formation of disulfides [76,77,78].

### 4.2. Phosphorylation of MMPs

Protein phosphorylation is one of the most important PTMs for regulation of biological processes [79]. This dynamic PTM is characterized by the reversible enzymatic addition of a phosphate group to amino acid side chains of serine (Ser), threonine (Thr), or tyrosine (Tyr), resulting in alterations of protein structure, stability, and dynamics [79]. Hence, phosphorylation can initiate different conformational changes associated with differential activity and binding specificity, leading to an activation or deactivation of proteins [80,81]. Furthermore, the dynamic nature of phosphorylation resulting from the opposing activities of kinases (which add phosphate groups) and phosphatases (which remove phosphate groups) is functionally compatible with the modulation of intramolecular interactions, implicated in essential cellular processes, such as cell division, cellular proliferation and differentiation, apoptosis, and signal transduction [79]. Consequently, dysregulation of phosphorylation dynamics is often related with pathologies, e.g., tumor formation, chronic inflammatory diseases, autoimmune diseases, and neurodegenerative disorders [81,82].

As many secreted and extracellular proteins, the MMPs are phosphorylated. Ser-, Thr-, and Tyr phosphorylation sites are identified across the different structural domains of the MMPs. Yet, the biological significance of these phosphate groups is overall poorly characterized. Among the MMPs, MMP2 and membrane-type MMP14 metalloproteinase are the ones with most extensively studied phosphorylation-dependent effects.

#### 4.2.1. MMP2

MMP2 contains 29 potential phosphorylation sites distributed across the pro-peptide domain, collagen-binding domain, collagenase-like domain-1 and -2, and hemopexin domain. However, only five of the predicted phosphorylation sites (S32, S160, Y271, T250, and S365) have been confirmed by mass spectrometry (MS). The phosphorylation of MMP2 noticeably diminishes its proteolytic activity, while dephosphorylation increases MMP2 activity. This is possibly due to the conformational changes observed in the secondary structure of dephosphorylated MMP2, with α-helices 50% longer and β-strands 17% shorter than phosphorylated MMP2. However, it is still unclear if protease activity is directly influenced by these conformational changes [83,84].

#### 4.2.2. MMP14

MMP-14 phosphorylation is known to play an important role in fine-tuning the activity of MMP14 at the cell surface and in the activation of intracellular signaling, and it is critical for regulating the pro-metastatic function of this metalloproteinase. This membrane-type metalloprotease has nine potential phosphorylation sites in its multidomain structure (PhosphoSitePlus [85], UniProt [86]). The cytoplasmic domain of MMP14 regulates its internalization and trafficking, thereby modulating enzymatic activity at the cell surface. A study by García-Pardo et al. reported that this domain was essential for MMP14 mediated cellular invasion and migration [72]. The cytoplasmic domain of MMP14 contains two phosphorylation sites, at Thr^567^ and Tyr^573^, which have a significant influence on MMP14-induced cellular invasion and migration. A study by Williams et al. in fibrosarcoma cells showed that substitution of the residue Thr^567^ by alanine increased retention at the cell surface and reduced internalization of the protease significantly compromising invasion and migration. Contrarily, mimicking protease phosphorylation by substitution of Thr^567^ by glutamic acid reduced retention of MMP14 at the cell surface, increased efficiency of internalization, and was correlated with an increase in migration and invasion [87]. A study by Moss et al. showed that phospho-mimetic Thr^567^ mutants exhibit higher collagenolytic activity and three-dimensional growth within a collagen matrix, thereby promoting enhanced matrix invasion in ovarian cancer cells [88]. In addition, phosphorylation of Thr^567^ impacted the integrity of cell monolayer, cell motility and multicellular aggregate dynamics in ovarian cancer cells, promoting metastasis-associated behaviors [89]. Furthermore, a study by Nyalendo et al. reported that phosphorylation at Tyr^573^ influenced cell migration, suggested by the ability of a phospho-defective mutant to inhibit migration of cells endogenously expressing MMP14 [90]. Additionally, epidermal growth factor (EGF)-induced phosphorylation of Tyr^573^ prompted internalization of MMP14 together with pericellular collagen, establishing an environment for expansive growth in three–dimensional collagen matrix, whereas lack of responding to EGFR signaling triggered invasive growth. Therefore, the phosphorylation of Tyr^573^ modulates cell surface dynamics of MMP14, thereby regulating the transition between invasive and expansive growth [91].

#### 4.2.3. Extracellular Phosphorylation of MMPs

A subset of MMPs (MMP1, 12, 13, 14, 16, 24, and 27) are tyrosine phosphorylated by the extracellular vertebrate lonesome kinase (VLK). These phosphotyrosines are found exclusively within the hemopexin domains of the MMPs. Interestingly, proteins distantly related to MMPs, which contain hemopexin-like domains, were found to be tyrosine phosphorylated at identical positions by VLK. The structural conservation of these phosphorylation sites suggests that they play a role in regulating MMP activity, a potential function, which still remains largely undiscovered [92,93]. Additionally, a study by Bordoli et al. showed that co-expression of VLK with MMPs supported their tyrosine phosphorylations that have been extensively observed in vivo. Bordoli also showed that by introducing a mutation in the ATP binding site of VLK or deletion of the proline glycine-rich domain (a conserved domain close to the kinase domain), the phosphorylation of the MMPs and other co-expressed substrates in the extracellular environment was eliminated. Similarly, shRNA-mediated downregulation of VLK expression and substitution of Tyr^360^ with phenylalanine in MMP1 reduced MMP13 and MMP1 tyrosine phosphorylation, respectively [93]. While VLK generates the majority of extracellular phosphotyrosines, additional secreted kinases have been identified, which may also be responsible for the extracellular protein phosphorylation events observed in vivo. This indicates a major impact of the secreted kinome on tissue homeostasis and disease pathogenesis. Discovery of novel extracellular kinases, identification of their substrates, and dissection of the regulatory mechanisms involved will provide us with a better understanding of their functions [94,95,96].

### 4.3. Glycosaminoglycans

An additional layer of regulation of extracellular proteolysis is mediated by interactions of MMPs with glycans in the extracellular space rather than direct glycosylation or phosphorylation of the protease or substrate. Glycosaminoglycans (GAGs) are extracellular glycans, which are composed of linear, unbranched repeats of disaccharide units. Heparan sulfate (HS) is a GAG-family member expressed by virtually every cell of a multicellular organism composed of repeats of glucoronic acid (GlcA) and *N*-acetylglucosamine (GlcNAc) dimers, which in most cases are attached to a cell-membrane associated core protein. During GAG-chain synthesis HS undergoes several sulfotransferase-mediated *N*- and *O*-sulfations. These modifications do not undergo completion, which results in strongly negatively charged sugar chains with variable length and degree of sulfation interrupted by stretches of unmodified, neutral regions. The sulfated regions bind basic peptide motifs of many signaling molecules, including morphogens (Hedgehogs, Wnts, TGFs), cytokines, and most chemokines [97,98,99]. This electrostatic interaction leads to partial neutralization of the protein surface and can have different effects on the biological functions of the interacting protein by facilitating, enhancing, or inhibiting the interaction with other proteins. It is important to note that these interactions, even though they are mainly based on electrostatics, are in most cases very specific and do not occur randomly. This is shown by the fact that many proteins contain specific GAG-binding motifs, with which they exclusively bind HS, even though they contain many lysine and arginine residues (for an in-depth review of the molecular basis of HS-protein interactions see Xu & Esko, 2014 [100]). Examples of HS-binding proteins are members of the fibroblast growth factor (FGF) family. Here, HS is part of the trimeric FGF/FGF-receptor/HS co-receptor complexes and activates FGF signaling in a sulfation pattern-dependent manner [101,102,103,104]. HS-induced oligomerization has been shown in many other processes including dimerization of amyloid precursor protein (APP) complexes or nearly all chemokines [105,106,107,108]. Interestingly, HS does not only mediate protein complex formation, but many proteins binding to HS, including the aforementioned candidates, are also known to undergo proteolytic processing on or near the cell surface. This indicates a crucial role of HS in regulating extracellular cleavage processes.

#### 4.3.1. GAG-regulated Substrate Proteolysis

Signaling processes in the extracellular space usually involve at least one form of substrate proteolysis. Therefore, co-factors have to act as decision makers to ensure regulated processing of signaling proteins. Multiple cases have been reported, where this is mediated by substrate binding to GAGs. For example, similar to *N*-glycans, GAGs can modulate the stability of substrates. FGF2 binding to heparin, an extremely sulfated HS variant expressed by connective-tissue type macrophages, stabilizes FGF2 and protects it from degradation by proteolysis [109,110]. The same stabilizing effect has been shown for the degradation of stromal cell-derived factor 1 (SDF1)/CXCL12 and C-terminal processing of interferon-γ [111,112]. Despite this stabilizing effect GAGs have been demonstrated to play additional important roles in extracellular proteolysis. In the case of shedding of morphogens from the Hedgehog (Hh) family, HS acts as an assembly platform for protease-release complexes [113]. Here, HS recruits Hh co-factors that co-localize proteases and allow for cleavage of the substrate (Figure 3) [114]. Furthermore, direct HS binding of the N-terminal cleavage site of Hh also has a stabilizing effect by inhibiting N-terminal Hh processing and reduces its release in vitro and in *Drosophila* in vivo [114,115]. An example of how GAG binding can positively regulate substrate cleavage (Figure 3) was shown for the cleavage of viral capsid proteins. Binding of the human papillomavirus viral capsid protein L1 to highly sulfated HS is essential to induce a conformational change and leads to cleavage of L1 by the human trypsin-like serine proteinase kallikrein 8, a process which is required for virus internalization [116].

#### 4.3.2. GAG-regulated MMP Activity

GAGs also regulate extracellular proteolysis at the protease level by modulating activation, localization, and protease–substrate interactions (Figure 3 middle). Many secreted vertebrate MMPs associate to the cell surface via binding to HS, including MMP9, which is specific, since excess soluble heparin extracts and solubilizes MMP2, MMP7, MMP9, and MMP13 [117,118]. Moreover, heparin, a highly sulfated form of HS, affects protease activity by increasing TIMP3 affinity to MMP2, MMP7, and MMP9, and HS affects MMP9 expression and plasma levels, depending on its sulfation and expressing cell type [119,120,121,122,123]. Finally, in vivo studies showed that MMPs in secretory granules of mast cells interact with heparin and that loss of heparin sulfation reduces MMP stability [124,125]. Like all MMPs, the wound repair and immune reaction associated MMP7 has to be converted into its proteolytic active form by removal of its inhibitory pro-sequence. MMP7 is able to activate itself by intermolecular autolytic processing. This is amplified by binding of MMP7 to highly sulfated GAGs (e.g., heparin or chondroitin sulfate), which significantly increases MMP7 auto-processing and also increases cleavage of specific physiological substrates [19].

## 5. Conclusions

MMP biology has been revolutionized with the recognition of extracellular proteolysis not as a simple mechanism of ECM degradation but as a regulatory mechanism for precise cellular control of biological processes. The paradigm shift for MMP functions from ECM degradative proteases to important regulators of essential cellular processes has highlighted the physiological relevance of these proteases, directly implicated by the relationship between MMP expression and disease development [4,8]. Hence, the multilayered regulation of MMPs emphasizes the tremendous importance of the balance between synthesis of active enzymes and their inhibition, which is pivotal to avoid the destructive activity of these proteases. Characterization of these regulatory mechanisms will aid the development of new therapeutics for various pathologies. MMP activity is regulated at the transcriptional level, post-translationally by pro-enzyme activation, by PTMs, and through extracellular inhibition by TIMPs and by non-specific proteinase inhibitors. Among these regulatory mechanisms, PTMs have recently obtained wide attention in the MMP community. MMPs are modified by PTMs at multiple sites (Table 1), which affects the activity of individual MMPs to a different extent, but the biological relevance of many of them is still unknown. However, many of the studies referenced in this review have been performed prior to the rapid progression of proteomics method development within the last decade. Many of the challenges that still limit our understanding of PTM function in MMP biology can now be approached by modern technologies of state-of-the-art proteomics. Advances in MS-based methods including multiplexed chemical labeling, novel label-free quantification strategies, improvement in PTM enrichment, more robust PTM analysis workflows and streamlined bioinformatics strategies will contribute to reliable identification and quantification of high numbers of PTMs [126,127]. The remaining challenge lies in defining the functional role of the physiologically relevant PTMs to understand PTM-dependent activity of MMPs in complex biological systems [128].

## Figures and Tables

**Figure 1 ijms-20-03077-f001:**
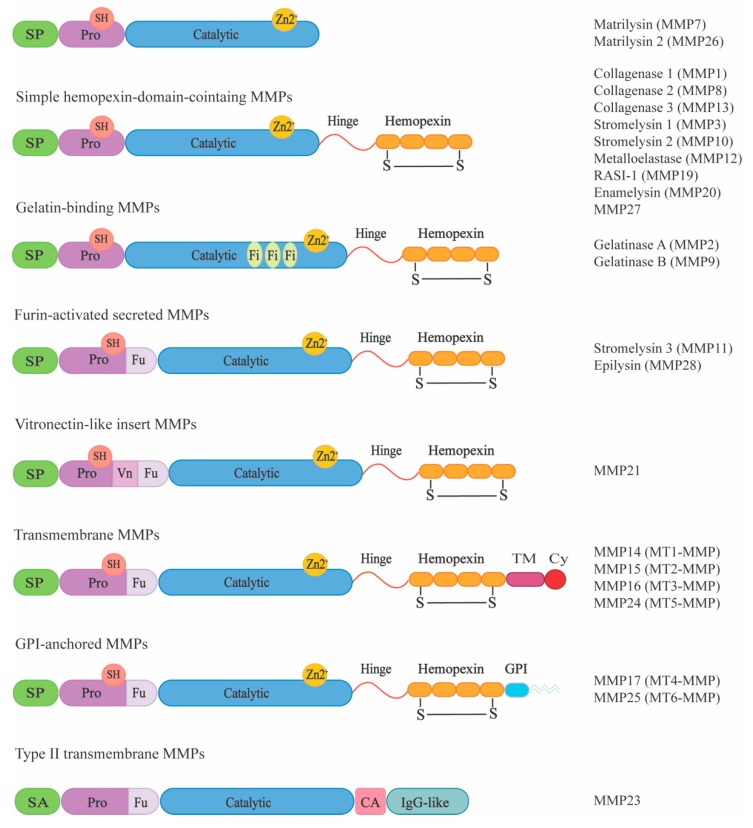
MMP multidomain organization. Based on their domain structure, MMPs can be subdivided into eight different groups. All MMPs share a common structure comprising a signal peptide (SP), a pro-domain (Pro), containing a thiol group (SH), a catalytic domain (Catalytic) with a zinc (Zn) binding site, a linker region (Hinge) and a hemopexin domain (Hemopexin), which has a disulfide bond (S-S). Exceptions to this are the two gelatinases, which contain three fibronectin repeats (Fi) within their catalytic domain and furin-activated MMPs which have a furin-recognition site (Fu) within their pro-domain and MMP21 with an additional vitronectin-like insert (Vn). Some membrane-type MMPs are anchored to the membrane via glycosylphosphatidylinositol (GPI), whereas some MT-MMPs have transmembrane (TM) and cytosolic domains (Cy). In type II MT-MMPs, an N-terminal signal anchor (SA), a cysteine array (CA) domain, and an immunoglobulin-like (Ig-like) domain are present. Adapted by permission from “Springer Nature: New functions for the matrix metalloproteinases in cancer progression. *Nat. Rev. Cancer*
**2002**, *2*, 161–174. Egeblad & Werb, Copyright (2002)”.

**Figure 2 ijms-20-03077-f002:**
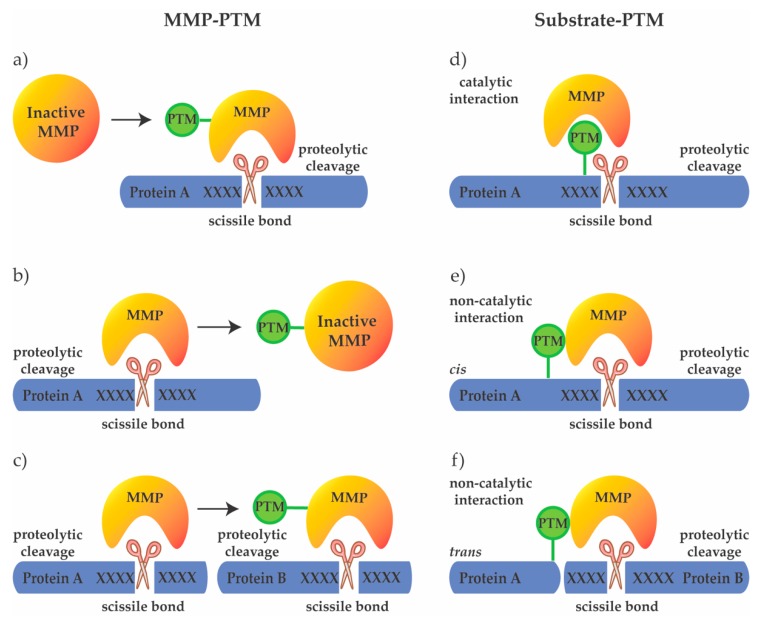
Possibilities of PTM crosstalk with MMPs. PTMs can regulate the activity of MMPs on the protease level by (**a**) activating the protease to allow cleavage of the target protein, (**b**) inactivating the protease, or (**c**) modulating its substrate specificity. Additionally, PTMs may regulate substrate cleavage by MMPs, through (**d**) directly modifying the cleavage motif (catalytic interaction), or substrate domains outside the cleavage site (non-catalytic interaction), guiding MMPs to cleave the same (*cis*) (**e**) or another (*trans*) protein (**f**).

**Figure 3 ijms-20-03077-f003:**
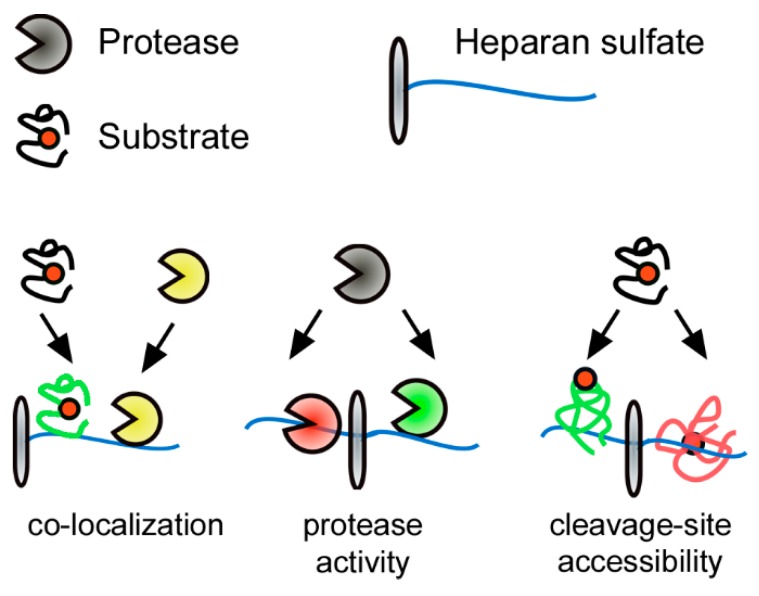
Heparan sulfate (HS)-protein interaction as a post translational regulator of extracellular proteolysis. Electrostatic interactions of proteins with negatively charged HS affect extracellular proteolysis at three different levels: (1) by mediating co-localization of reactants; (2) on the protease level by affecting protease activity/accessibility; or (3) on the substrate level by modulating accessibility to cleavage sites. The substrate cleavage site is indicated by a red circle, green substrate/protease color denotes activity; red substrate/protease color denotes inactivity/inaccessibility.

**Table 1 ijms-20-03077-t001:** List of identified PTMs in MMPs.

MMP	Modification	Biological Effect	Reference
MMP1	*N*-linked glycosylation at Asn^120^	Tumor cell invasion and angiogenesis	[39,69,129]
	Phosphorylation at Tyr^360^	Not reported	[93]
MMP2	*O*-linked glycosylation at Ser^32^, Thr^96, 262, 458, 460^	Upregulation of MMP2	[39,71,72]
	*N*-linked glycosylation at Asn^573^ and Asn^642^	Not reported	[130]
	Phosphorylation at Ser^32^, Ser^160^, Tyr^271^, Thr^250^ and Ser^365^	Phosphorylation decreases, while dephosphorylation increases protease activity	[83,84]
	Heparan sulfate	Cell surface localization; affects protease activity by increasing TIMP3 affinity	[119,120,121,122]
MMP3	*N*-linked glycosylation at Asn^120^ and Asn^398^	Not reported	[39,73]
	Three potential *O*-linked glycosylation at Ser^56^, Ser^269^ and Thr^277^		
MMP7	Heparan sulfate, Chondroitin sulfate	Cell surface localization; affects protease activity by increasing TIMP3 affinity; increases MMP7 auto-processing and activity	[19,120,121]
MMP9	*N*-linked glycosylation at Asn^38^ and Asn^120^	MMP9 secretion and activation	[53,54,55,56,57,58,59]
	*O*-linked glycosylation in the linker region	Increases the domain flexibility; necessary for internalization and degradation; protects against proteolytic degradation; reduces gelatinolytic activity	[60,62,63,64]
	Heparan sulfate	Cell surface localization; affects protease activity by increasing TIMP3 affinity; affects MMP9 expression and plasma levels	[119,120,121,122,123]
MMP12	Phosphorylation at Tyr^414^	Not reported	[93]
MMP13	*N*-linked glycosylation at Asn^117^ and Asn^152^	Not reported	[1,74,75]
	*O*-linked glycosylation at Ser^24^ and Ser^62^	Not reported	[1,74,75]
	Phosphorylation at Tyr^366^	Not reported	[93]
MMP14	*N*-linked glycosylation at Asn^229^ and Asn^311^	Not reported	[39]
	*O*-linked glycosylation at Thr^291^, Thr^299^, Thr^300^, and Ser^301^	Required for formation of a stable complex with proMMP2 and TIMP2; increases activity upon glycosylation perturbation	[65,66,67,68]
	Phosphorylation at Thr^567^, Tyr^573^ and Tyr^353^	Regulates MMP14 induced cellular invasion and migration; cell surface dynamics and internalization; mimetic mutants exhibit higher collagenolytic activity and three-dimensional growth; promotes metastasis-associated behaviors	[72,87,88,89,90,91,93]
MMP16	Phosphorylation at Tyr^377^ and Tyr^521^	Not reported	[93]
MMP17	*N*-linked glycosylation at Asn^137^ and Asn^318^	Stabilizes the dimeric form of MMP17	[76,77,78]
MMP24	Phosphorylation at Tyr^534^	Not reported	[93]
MMP27	Phosphorylation at Tyr^360^	Not reported	[93]

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
