# Peer review of "Post-Translational Modification-Dependent Activity of Matrix Metalloproteinases"

_ijms, 2019, doi:10.3390/ijms20123077_

Round 1

Reviewer 1 Report

The authors reviewed the post-translational modifications-dependent activity of matrix metalloproteinases. The study purpose is quite interesting but the data analysis and organization are weak. So, the major decision was required as the following reasons.

1.     Line 24-25. The number of MMPs members are different between human and mouse. Please write in details.

2.     Line 102-103. Please add the information of PTMs classification due to their properties, structures, and functions.

3.     Line 115-116. Glycosylation of MMPs

4.     Section 4.1. The information of MMP 1, 9, … from subsections of section 4.1. have been clearly shown in other reviews. Please find the appropriate citations. If the authors still want to add that information, please make a table.

4.2.1.     Phosphorylation of MMPs

4.2.2.     Glycosaminoglycans

4.2.3.     The mechanism of PTM-MMPs or MMPs’ substrates relationship should be described in sections 4.1-4.3 in more details.

5.     Please add the figures describing the interaction of MMPs or MMPs’ substrates and PTMs.

6.     Please refer some more publications related to your topics such as PMID 29903573, 30443247, 28972180, 24955328 and 23887885.

Minor comments: There are some pieces of important information existed the lack of citations. Please check again which parts are more important and needed to have citations.

Author Response

We thank the reviewer for the thorough and critical inspection of our manuscript. We have carefully evaluated all comments and made major revisions including incorporating additional illustrations and tables to improve our review article. Please find point-by-point responses below.

1.     Line 24-25. The number of MMPs members are different between human and mouse. Please write in details.

Text and Figure 1 refer to human MMPs. We have now added this information specifically to line 25.

2.     Line 102-103. Please add the information of PTMs classification due to their properties, structures, and functions.

PTM categorization is provided in lines 98-107.

3.     Line 115-116. Glycosylation of MMPs

In this line the general phenomenon of glycosylation is introduced, which does not only refer to MMPs.

4.     Section 4.1. The information of MMP 1, 9, … from subsections of section 4.1. have been clearly shown in other reviews. Please find the appropriate citations. If the authors still want to add that information, please make a table.

All appropriate citations have been added and in addition summarized in a new Table 1.

4.2.1.     Phosphorylation of MMPs

See comment on line 115-116.

4.2.2.     Glycosaminoglycans

4.2.3.     The mechanism of PTM-MMPs or MMPs’ substrates relationship should be described in sections 4.1-4.3 in more details.

To better illustrate this concept, we have now added a new Figure 2.

5.     Please add the figures describing the interaction of MMPs or MMPs’ substrates and PTMs.

See comment to 4.2.3.

6.     Please refer some more publications related to your topics such as PMID 29903573, 30443247, 28972180, 24955328 and 23887885.

References have been added as suggested.

Minor comments: There are some pieces of important information existed the lack of citations. Please check again which parts are more important and needed to have citations.

We carefully checked the manuscript and included additional citations increasing the number of references from 124 to 130.

Reviewer 2 Report

This review entitled “Post-translational modifications-dependent activity of matrix metalloproteinases” describes the recent progress towards understanding the role of post-translational modifications (PTMs) such as glycosylation, phosphorylation, glycosaminoglycans on the activity of several members of the MMP family. Authors detail an overview about MMP domain structure and classification, MMP substrates and function, Multilayered regulation of MMP activity and PTMs – an additional level of protein regulation for easy comprehension when reading.

Author Response

We thank the reviewer for the positive comments on our manuscript.

Round 2

Reviewer 1 Report

The new version of the manuscript is acceptable for publication.